# Cardiac Autonomic Dysfunction Is Associated with Severity of REM Sleep without Atonia in Isolated REM Sleep Behavior Disorder

**DOI:** 10.3390/jcm10225414

**Published:** 2021-11-19

**Authors:** Sooyeoun You, Kyoung Sook Won, Keun Tae Kim, Hyang Woon Lee, Yong Won Cho

**Affiliations:** 1Department of Neurology, Keimyung University School of Medicine, Daegu 42601, Korea; omoi81@dsmc.or.kr (S.Y.); 6k5upa@gmail.com (K.T.K.); 2Department of Nuclear Medicine, Keimyung University School of Medicine, Daegu 42601, Korea; won@dsmc.or.kr; 3Departments of Neurology, Medical Science, Computational Medicine, System Health Science & Engineering, Ewha Womans University School of Medicine and Ewha Medical Research Institute, Seoul 07985, Korea

**Keywords:** REM sleep behavior disorder (RBD), sleep, autonomic dysfunction, ^123^I-metaiodobenzylguanidine (MIBG) scintigraphy, REM sleep without atonia

## Abstract

^123^I-metaiodobenzylguanidine (MIBG) cardiac scintigraphy was performed to assess cardiac autonomic dysfunction and demonstrate its correlation with clinical and polysomnographic characteristics in patients with isolated rapid eye movement (REM) sleep behavior disorder. All subjects including 39 patients with isolated REM sleep behavior disorder and 17 healthy controls underwent MIBG cardiac scintigraphy for cardiac autonomic dysfunction assessment. The isolated REM sleep behavior disorder was confirmed by in-lab overnight polysomnography. A receiver operating curve was constructed to determine the cut-off value of the early and delayed heart-to-mediastinum ratio in patients with isolated REM sleep behavior disorder. Based on each cut-off value, a comparison analysis of REM sleep without atonia was performed by dividing isolated REM sleep behavior disorder patients into two groups. MIBG uptake below the cut-off value was associated with higher REM sleep without atonia. The lower heart-to-mediastinum ratio had significantly higher REM sleep without atonia (%), both with cut-off values of early (11.0 ± 5.6 vs. 29.3 ± 23.2%, *p* = 0.018) and delayed heart-to-mediastinum ratio (9.1 ± 4.3 vs. 30.0 ± 22.9%, *p* = 0.011). These findings indicate that reduced MIBG uptake is associated with higher REM sleep without atonia in isolated REM sleep behavior disorder.

## 1. Introduction

Rapid eye movement (REM) sleep behavior disorder (RBD) is a parasomnia characterized by dream enactment behavior with vivid dreams and REM sleep without atonia (RWA) [1]. The disease was first reported in 1986 by Schenck et al. [2]. There exists a well-established concept to associate RBD with neurodegenerative diseases caused by alpha-synucleinopathy, such as Parkinson’s disease (PD), multiple system atrophy (MSA), and dementia with Lewy bodies (DLB). In previous studies, more than 50% of patients with RBD had parkinsonian disorders within a decade; between 81 and 90% of patients with RBD develop a neurodegenerative disorder [3,4,5,6]. Accordingly, recent studies have suggested that ‘isolated RBD (iRBD)’ reflects the early phase of later development of neurodegenerative changes, as the concept of ‘prodromal parkinsonism’ has been suggested to represent an early expression of α-synucleinopathy [7]. According to the results of recently published prospective studies, the phenoconversion rate from iRBD to parkinsonism or dementia are similar [8,9].

As interest in ‘disease-modifying therapy’ for neurodegenerative disease has increased, the identification of prodromal or pre-motor stages of PD has also been an area of interest [10].

However, markers predictive of the phenoconversion patients with RBD to neurodegenerative disease have not yet been accurately established. Therefore, there is a growing need for early biomarkers to identify and predict the disease course in patients with RBD who could progress to developing neurodegenerative diseases such as PD [11].

^123^I-MIBG myocardial scintigraphy is a non-invasive nuclear medicine test used to measure myocardial sympathetic nervous system function. It is also utilized for the differential diagnosis of neurodegenerative diseases such as PD and MSA [12,13,14]. It has been mentioned as a supportive tool in the Movement Disorders Society (MDS) diagnostic criteria of PD published in 2015 [15]. In PD, the heart-to-mediastinum ratio (HMR) of MIBG cardiac scintigraphy is reduced, suggesting a degenerative change in the postganglionic sympathetic nerve endings in the heart [16]. This stands in contrast with normal MIBG uptake, which is a characteristic of MSA [17,18]. However, some studies have shown that a decline in MIBG uptake can be observed in some MSA patients [19,20]. Reduced MIBG uptake is used as an indicative marker even in the diagnosis of DLB [21].

In previous studies related to MIBG cardiac scintigraphy, MIBG uptake was reduced in patients with iRBD [22,23,24,25,26]. MIBG uptake was abnormal in patients with prodromal symptoms of PD, such as autonomic dysfunctions and hyposmia [27]. In light of these findings, the use of MIBG as an early diagnostic method for PD may be taken into consideration. MIBG myocardial scintigraphy is useful because it can produce consistent and quantitative results in patients without underlying cardiac disease. 

The main aims of our study were (1) to confirm the difference in MIBG scintigraphy results between the normal and iRBD groups, (2) to analyze whether there is a correlation between the MIBG scintigraphy results and clinical and PSG data within the iRBD group, and (3) to find out the correlation between REM sleep without atonia as a main polysomnograhic feature of RBD and MIBG scintigraphy results. Moreover, we analyzed the MIBG uptake and clinical data to determine the clinical usefulness of MIBG cardiac scintigraphy as an early marker of neurodegenerative disease.

## 2. Materials and Methods

This is a prospective study, performed between May 2018 and October 2019.

This study was approved by the Institutional Review Board (IRB) of Keimyung University Dongsan Hospital (IRB No. 2018-05-018). All subjects provided written informed consent before enrolling in this study, in accordance with the Declaration of Helsinki.

### 2.1. Subjects

In this study, all 157 patients were confirmed to have RBD by overnight video polysomnography (PSG) and clinical interviews according to the 3rd edition of the International Classification of Sleep Disorders (ICSD-3) [1]. Patients with any medical history of neurodegenerative disease or the Korean version of the Mini-Mental Status Examination (K-MMSE) score under 21 were excluded. Epilepsy and sleep disorders that can mimic iRBD (e.g., moderate to severe obstructive sleep apnea (OSA) with Apnoea–Hypopnoea Index (AHI) > 15, restless legs syndrome, periodic limb movement disorder, and other parasomnias) were also excluded. In addition, all patients with major medical diseases that could affect the results of MIBG cardiac scintigraphy were excluded. These included heart failure, arrhythmia, coronary artery disease, hypertrophic and dilated cardiomyopathy, and drug-induced cardiomyopathy, diabetes mellitus, or a history of medication or substance use that could affect MIBG uptake results (e.g., reserpine, labetalol, or sympathomimetics such as tyramine, tricyclic antidepressants, and cocaine). As a result, 39 iRBD patients who did not meet the exclusion criteria and agreed to participate in the study were enrolled. These exclusion criteria were also applied to the recruitment of age-matched healthy volunteers to gather 17 age-matched controls, recruited after detailed clinical interviews as well as neurologic examinations to confirm that there are no clinical abnormalities related to sleep and autonomic dysfunction suspected (Figure 1).

### 2.2. Clinical Assessments

The Unified Parkinson’s Disease Rating Scale Part III (UPDRS-III), and the Hoehn and Yahr (H&Y) stages were used to assess the motor symptoms of parkinsonism. After an in-person face-to-face interview and physical examination, all subjects were asked to complete a series of questionnaires, including the Korean-validated versions of the Beck Depression Index (BDI) [28], Beck Anxiety Index (BAI) [29], Insomnia Severity Index (ISI) [30], Epworth Sleepiness Scale (ESS) [31], Pittsburgh Sleep Quality Index (PSQI) [32], RBD questionnaire (RBDQ-KR; cut-off value for total scoring is 18.5) [33], Scales for Outcomes in Parkinson’s Disease for Autonomic Symptoms (SCOPA-AUT) [34], the Korean version of the Sniffin’ stick (KVSS) test for olfactory function [35] and the K-MMSE. Neurological examination was performed by a movement specialist.

### 2.3. Polysomnographic Recording

All patients with iRBD underwent overnight PSG recording in a sleep laboratory. The electromyography (EMG) activity to detect RWA was assessed in the mentalis muscle [36]. The PSG data were analyzed according to the criteria described in the American Academy of Sleep Medicine (AASM) manual [37]. The quantification of REM sleep without atonia (RWA) was performed manually through a review of PSG records. Tonic activity was defined as an increase of >50% in baseline EMG amplitude (measured during non-REM sleep) or >10 mV. We obtained the ratio of time with tonic activities out of the total REM sleep time, and the ratio of the number of epochs showing tonic activity among the total number of REM sleep episodes, respectively. Phasic activity was defined as the time when a single epoch divided by 10 is divided into a mini epoch in 3-second increments, and when there were 5 or more non-contiguous mini epochs in which the EMG findings with an amplitude increase of more than 4 times lasted for 0.1 to 5 s. Depending on the dominant case (more than 50%) in one epoch, the tonic or phasic EMG activity was scored [36]. We calculated the ratio of the period in which phasic activity appeared during the total REM sleep time and the ratio of the number of mini epochs showing phasic activity among the total number of mini epochs in REM sleep.

### 2.4. ^123^I-Metaiodobenzylguanidine Cardiac Scintigraphy

All subjects in this study underwent MIBG cardiac scintigraphy with intravenous ^123^I-MIBG 111 MBq (3 mCi). Early and delayed images were obtained after 15 min and 3 h, respectively. For MIBG cardiac scintigraphy, all images were acquired using a Signature E.CAM Dual camera (Siemens Healthcare) with a medium-energy collimator. At the same time, gated myocardial perfusion single-photon emission computed tomography (SPECT) with ^99m^Tc-MIBI (methoxy isobutyl isonitrile) at rest was also performed with a D-SPECT camera (Spectrum Dynamics). Image processing and calculation of early and delayed HMR were performed by a nuclear medicine specialist. Whole MIBG uptake was assessed using the early and delayed images by manually drawing the region of interest (ROI) over the whole heart and mediastinum [38]. Gated myocardial perfusion SPECT with ^99m^Tc-MIBI at rest was used to distinguish asymptomatic myocardial infarction or cardiomyopathy.

### 2.5. Statistical Analysis

Statistical analyses were performed using SPSS Statistics (version 25.0, IBM Corp., Armonk, NY, USA). Statistical significance was set to *p*  <  0.05. The Mann–Whitney U-test and Fisher’s exact test were used for the statistical comparisons of demographics and, clinical and MIBG data between the iRBD and healthy control groups. The sensitivity, specificity, positive predictive value, and negative predictive value of each parameter from MIBG scintigraphy (early and delayed HMR) were analyzed by the SPSS software; each threshold had a 95% confidence interval (CI). A receiver operating curve (ROC) analysis was performed to determine the cut-off value of early and delayed HMR in patients with iRBD. Based on each cut-off value, a data comparison analysis was performed by dividing patients with iRBD into two groups. Spearman correlation analyses were performed between the results of MIBG cardiac scintigraphy, and the clinical and PSG data of patients with iRBD.

## 3. Results

In total, 39 patients with iRBD (25 men, 64.1%) and 17 healthy controls (6 men, 35.3%) were enrolled in this study (Table 1). The mean age of the patients with iRBD and healthy controls were 65.4 ± 6.5 and 66.0 ± 7.4, respectively. The ratio of males in the patients with iRBD was significantly higher than that among healthy controls (*p* = 0.044), and the patients with iRBD had higher scores of RBDQ-KR and gastrointestinal subscore of the SCOPA-AUT. However, there was no statistically significant difference in the total SCOPA-AUT and KVSS scores between the two groups.

Gated myocardial perfusion scans revealed normal perfusion and left ventricle function in all subjects. This confirmed that there was no underlying heart disease that could cause MIBG uptake abnormalities in all subjects. Comparison of MIBG cardiac scintigraphy results between patients with iRBD and healthy controls revealed statistically significant differences in early and delayed HMR. Both HMR results were obtained at an early (1.8 ± 0.3 vs. 2.7 ± 0.4, *p* < 0.001) and delayed time point (1.6 ± 0.5 vs. 2.8 ± 0.6, *p* < 0.001) and were lower in patients with iRBD than in healthy controls (Table 1 and Figure 2).

PSG data for the 39 patients with iRBD who underwent MIBG myocardial scintigraphy is summarized in Appendix A.

Receiver operating characteristic (ROC) analysis of the early and delayed HMR showed an area under the curve of 0.95 (95%, CI = 0.90–1.00) and 0.94 (95%, CI = 0.88–1.00) for iRBD (Figure 3). An early HMR with a cut-off value of 2.15 revealed a sensitivity and specificity of 100% and 87.2% in the differentiation between patients with iRBD and healthy controls, and a delayed HMR with a cut-off value of 2.05 revealed a sensitivity and specificity of 88.2% and 89.7%, respectively.

According to the cut-off value, patients with iRBD can be divided into two groups. The comparison between patients with iRBD is shown in Table 2 and Table 3. The lower HMR had significantly higher RWA (%), both with a cut-off value reflecting early (Table 2) and delayed HMR (Table 3). However, RWA revealed no significant correlation with early (*p* = 0.604) or delayed HMR (*p* = 0.806) (Appendix A).

## 4. Discussion

In this study, we calculated the cut-off value of RWA severity and divided the study population it into two groups based on it. Accordingly, when MIBG scintigraphy was compared, a significant difference between the two groups was observed.

Since RBD is known to correspond to a prodromal stage of alpha-synucleinopathy-related neurodegenerative disease, an objective marker of iRBD may suggest or predict the presence of alpha-synucleinopathy. In this study, we identified a decrease in MIBG uptake in patients with iRBD compared to healthy controls, and found that iRBD subjects with a low MIBG uptake had higher mean RSWA percentages on the PSG. Previous studies have reported a decrease in MIBG uptake in patients with iRBD [22,23,27,39]. The pattern of MIBG uptake in RBD patients is analogous to that of PD/DLB [40]. Therefore, a decrease in MIBG uptake before the occurrence of presynaptic dopaminergic neuronal loss indicates that MIBG cardiac scintigraphy may be a biomarker. However, there remains insufficient evidence to determine whether reduced MIBG absorption in patients with RBD predicts conversion to PD or suggests a worse outcome.

A previous study reported that the cognitive functions of patients with RBD may deteriorate following the diagnosis of PD, not prior to their conversion to PD [41]. However, decreased MIBG uptake in cardiac scintigraphy, which represents the sympathetic denervation of the heart, is profound in PD with RBD, PD with orthostatic hypotension, and PD with wearing off [42,43]. In addition, low MIBG uptake predicts Lewy body pathology prior to the motor symptoms of Parkinsonism [24,44]. This suggests that the decreased uptake in MIBG cardiac scintigraphy may be an indicator of diffuse distribution in alpha-synucleinopathy [45].

Several studies have suggested a cut-off value of MIBG uptake value for the differentiation between iRBD and normal controls, PD, or OSA [23,46,47]. RWA is associated with increased rigidity, progression of Lewy body dementia, and PD development [48,49,50]. Consequently, recent studies have indicated that RWA may serve as a biomarker for neurodegenerative diseases emerging from iRBD [51,52].

In this study, we set the cut-off value of RWA within the iRBD group and observed the difference in MIBG uptake according to the RWA results. This points to the clinical applicability of MIBG cardiac scintigraphy in predicting the diagnosis or diffuse distribution of alpha-synucleinopathy in iRBD patients. The early ^123^I-MIBG uptake mainly reflects the distribution of cardiac sympathetic nerves and the uptake function at the nerve endings, while the ^123^I-MIBG wash out primarily reflects synaptic vesicle release due to sympathetic tone. The early ^123^I-MIBG uptake mainly reflects the distribution of cardiac sympathetic nerves and the uptake function at the nerve endings, while the ^123^I-MIBG wash out primarily reflects synaptic vesicle release due to sympathetic tone. Therefore, low values of the delayed HMR are suggestive of structural (cardiac sympathetic innervation) and functional (increased cardiac sympathetic tone) alterations [53]. It remains unclear why differences in MIBG uptake were observed when the two groups were compared according to RWA severity. It has been suggested that medullary magnocellular reticular formation is related to the mechanism of RWA in RBD [54]. In addition, the ventromedial medulla, tegmental periaqueductal gray matter, and mesopontine cholinergic nuclei are also suggested to be related to RBD [55,56]. Since these brain stem structures are close to the sympathetic nerve activation center, such as nucleus tractus solitaries, a connection between them is highly probable. However, since MIBG scintigraphy mainly reflects postganglionic sympathetic denervation [57], brainstem pathology alone would not be able to explain the study results completely. Rather, the results may be a reflection the diffuse distribution of alpha-synucleinopathy in RBD [45].

Although this study failed to show statistically significant associations, we were able to observe variation in the HMR according to the RWA severity. We surmise that the widely distributed RWA (%) might have resulted in non-significance. In other words, the significance of this study is that we found the clinical significance of MIBG cardiac scintigraphy based on the cut-off value, although we could not show a statistical correlation between RWA and MIBG uptake due to the sharp changes in RWA.

In the subgroup analysis of patients with iRBD, when comparing the total, factor 1 (dream-related), and factor 2 (behavioral manifestation) scores of RBDQ-KR, there was no difference between the two groups. This means that changes in HMR precede differences in RBD symptoms, such as dream-related or behavioral manifestations. Although statistical significance was not observed, in the group in which HMR was higher than the cut-off value, the RBD onset age was younger, and the disease duration and interval of symptoms to PSG were longer. Even if the age of MIBG cardiac scintigraphy was similar, it can be considered that postganglionic heart sympathetic nerve endings are better preserved in patients whose RBD symptoms begin at a younger age. Although advanced age is known as a risk factor for PD incidence in patients with iRBD [58] no research has been reported on RBD onset age.

This study has several limitations in this study. First, it consisted of a small number of subjects from a single hospital. In this single center study, age-/sex-matched health controls could not be obtained, due to limitations in participant recruitment. Second, we cannot exclude secondary RBD caused by structural brain lesions. No additional imaging tests (such as DaTscans) were performed to differentiate iRBD from premotor stage PD. Contrarily, selection bias may have occurred due to the strict exclusion criteria for other medical conditions and sleep disorders. Third, no quantified autonomic function test was performed in all subjects, and PSG was not performed in the healthy control group. Only the EMG activity of the mentalis muscle was measured when performing the PSG test in RBD patients. To increase the sensitivity of the diagnosis, the addition of EMG leads for limbs is being considered in future studies [59]. We analyzed both phasic and tonic activities, as it is known that phasic EMG activity is always increased in iRBD, whereas tonic EMG activity may not be increased in some patients with RBD [60] which may have served as a limitation for the RWA analysis. Fourth, when a sub-analysis was performed within the iRBD group, there was a large numerical difference between the two subgroups. Additionally, we encountered difficulties in a correction for multiple comparisons due to the large number of variables. Lastly, this study would have been improved had it included a comparative group of neurological degenerative diseases.

## 5. Conclusions

This study revealed that decreased uptake of MIBG cardiac scintigraphy in patients with iRBD was associated with the severity of RWA. A long-term follow-up study or prospective study is needed to obtain clearer evidence on whether the combination of MIBG cardiac scintigraphy and RWA can be a potential biomarker predicting iRBD phenoconversion.

## Figures and Tables

**Figure 1 jcm-10-05414-f001:**
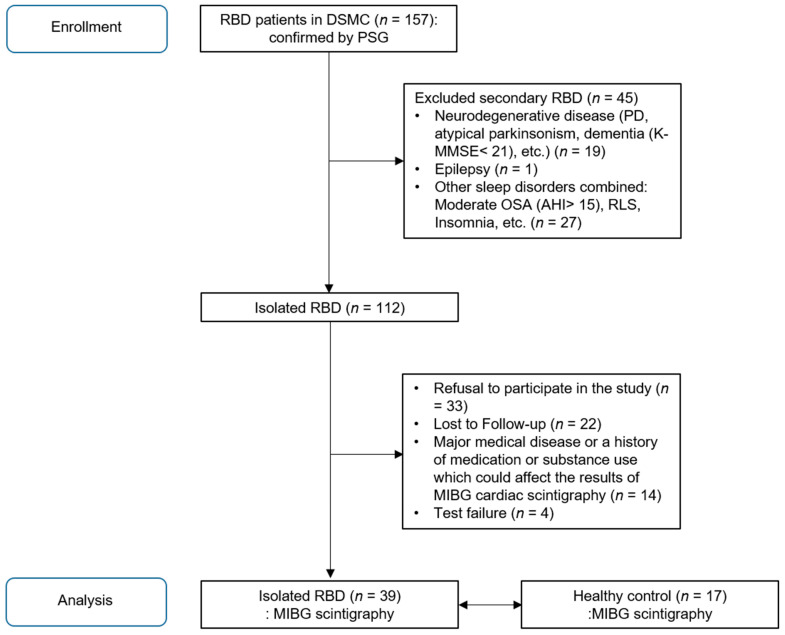
Recruitment process of idiopathic REM sleep behavior disorder patients for MIBG myocardial scintigraphy study.

**Figure 2 jcm-10-05414-f002:**
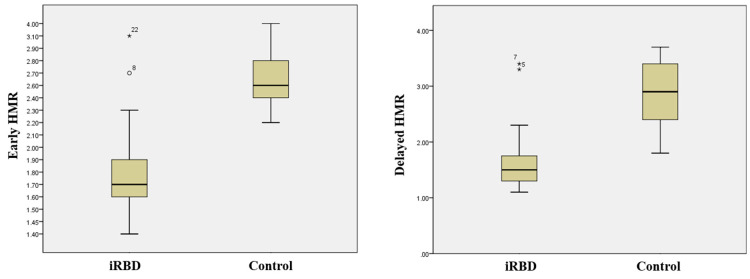
Distribution of the early and delayed HMR in control subjects and patients with iRBD. The horizontal lines indicate the median values. The early (15 min) (**left**) and delayed (3 h) (**right**) HMR are decreased in patients with iRBD. * *p* < 0.05.

**Figure 3 jcm-10-05414-f003:**
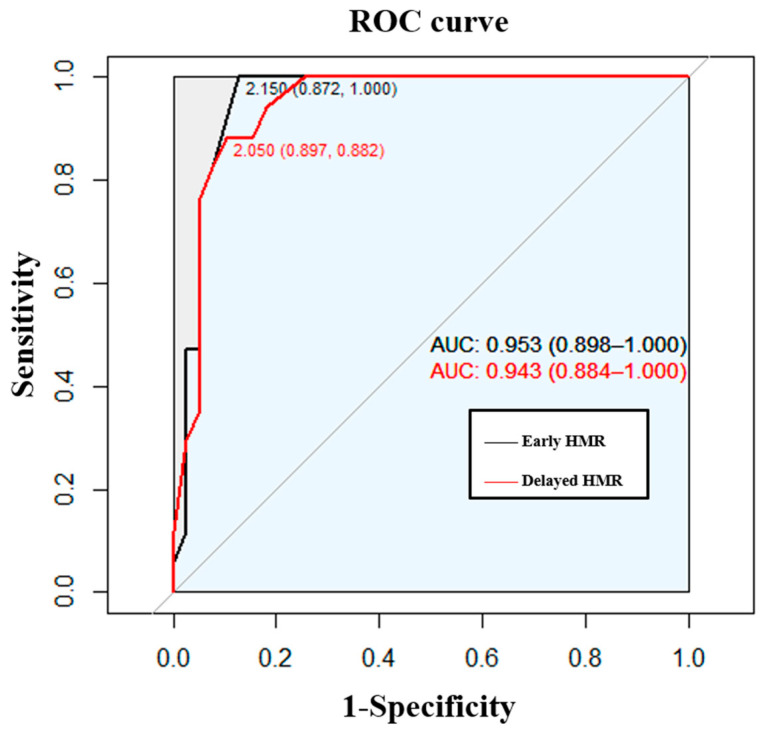
ROC analysis of the early or delayed HMR for patients with iRBD.

**Table 1 jcm-10-05414-t001:** Demographic features and clinical data of patients with iRBD and control subjects.

	iRBD (*n* = 39)	Healthy Control (*n* = 17)	*p*-Value
Age	65.4 ± 6.5	66.0 ± 7.4	0.954
Sex (male, %)	25 (64.1)	6 (35.3)	0.044 ^#^ *
RBD disease duration	6.5 ± 4.8 (0.25–13.0)	-	-
(range, years)			
H & Y stage	0	0	1
UPDRS part III	0.3 ± 1.0	0.6 ± 1.5	0.258
Sleep questionnaire			
ISI	6.1 ± 5.8	7.1 ± 7.2	0.961
ESS	4.8 ± 4.2	2.8 ± 2.8	0.065
PSQI	6.5 ± 3.6	6.2 ± 7.3	0.349
RBDQ-KR	42.8 ± 18.2	14.7 ± 19.4	<0.001 *
BDI	12.0 ± 10.8	11.5 ± 9.2	0.842
BAI	7.2 ± 7.9	6.3 ± 7.3	0.828
MMSE	27.1 ± 2.4	26.1 ± 2.8	0.184
SCOPA-AUT, total	8.9 ± 4.7	6.7 ± 5.3	0.054
Gastrointestinal	2.6 ± 2.1	1.2 ± 1.4	0.014 *
Urinary	4.7 ± 2.9	4.1 ± 3.0	0.415
Cardiovascular	0.5 ± 0.8	0.7 ± 1.3	0.911
Pupilomotor	0.4 ± 0.7	0.2 ± 0.7	0.293
Thermoregulatory	0.4 ± 0.9	0.2 ± 0.4	1
Sexual	0.4 ± 0.9	0.2 ± 0.8	0.279
KVSS test	4.2 ± 2.8	5.5 ± 2.2	0.185
Early HMR	1.8 ± 0.3	2.7 ± 0.4	<0.001 *
Delayed HMR	1.6 ± 0.5	2.8 ± 0.6	<0.001 *

Data are mean ± SD or *n* (%) values. # Fisher’s exact test. * *p* < 0.05. RBD: REM sleep behavior disorders, UPDRS: Unified Parkinson’s Disease Rating Scale, H & Y: Hoehn and Yahr, ISI: Insomnia Severity Index, ESS: Epworth Sleepiness Scale PSQI: Pittsburgh Sleep Quality Index, RBDQ-KR: RBD Questionnaire—Korean version, BDI: Beck Depression Index, BAI: Beck Anxiety Index, MMSE: Mini-Mental Status Examination, SCOPA-AUT: Scales for Outcomes in Parkinson’s Disease for Autonomic Symptoms, KVSS: Korean Version of Sniffin’ Sticks Test.

**Table 2 jcm-10-05414-t002:** Comparative analysis within iRBD group based on the cut-off value of early HMR.

	≥2.15 (*n* = 5)	<2.15 (*n* = 34)	*p*-Value
Age	63.6 ± 7.9	65.6 ± 6.3	0.752
Sex (male, %)	3 (60.0)	22 (64.7)	1.000 ^#^
UPDRS part III	0.8 ± 1.8	0.2 ± 0.9	0.243
Onset age, year	54.6 ± 2.6	60.2 ± 7.0	0.269
Duration, year Interval of symptoms to PSG, year	11.6 ± 10.7 9.8 ± 4.4	5.7 ± 2.8 3.0 ± 0.5	0.304 0.186
ISI	6.4 ± 6.9	6.0 ± 5.7	0.909
ESS	2.2 ± 2.2	5.2 ± 4.4	0.120
BDI	9.2 ± 10.6	12.4 ± 11.0	0.646
BAI	5.2 ± 7.1	7.4 ± 8.1	0.532
PSQI	6.4 ± 3.0	6.6 ± 3.3	0.925
MMSE	26.2 ± 3.1	27.2 ± 2.3	0.527
RBDQ-KR Factor 1 Factor 2	35.4 ± 18.5 10.6 ± 7.4 25.2 ± 17.1	43.9 ± 18.2 13.3 ± 5.7 31.0 ± 14.6	0.467 0.360 0.424
SCOPA-AUT total	6.8 ± 4.3	9.2 ± 4.7	0.289
Gastrointestinal	2.4 ± 1.7	2.6 ± 2.1	0.963
Urinary	3.4 ± 3.1	4.9 ± 2.9	0.248
Cardiovascular	0.2 ± 0.5	0.6 ± 0.8	0.483
Pupilomotor	0.4 ± 0.6	0.4 ± 0.7	0.609
Thermoreulatory	0.2 ± 0.5	0.3 ± 0.5	1.000
Sexual	0.2 ± 0.5	0.5 ± 0.9	0.794
KVSS test	4.6 ± 3.1	4.1 ± 2.7	0.745
MIBG			
Early HMR	2.5 ± 0.4	1.7 ± 0.2	<0.001 *
Delayed HMR	2.6 ± 0.7	1.5 ± 0.2	<0.001 *
Total sleep time (min)	355.3 ± 80.8	384.8 ± 55.4	0.522
N1 (%)	13.9 ± 4.9	13.8 ± 6.1	0.669
N2 (%)	47.6 ± 10.1	50.0 ± 8.4	0.419
N3 (%)	20.1 ± 3.4	15.0 ± 9.1	0.205
REM (%)	18.4 ± 6.3	21.4 ± 7.4	0.358
Latency to sleep onset	48.4 ± 84.1	12.6 ± 13.2	0.592
Latency to sleep stage 2 (min)	55.7 ± 82.8	18.1 ± 15.5	0.481
Latency to REM sleep stage (min)	110.1 ± 87.5	118.5 ± 70.0	0.967
RWA (%)	11.0 ± 5.6	29.3 ± 23.2	0.018 *
Sleep efficiency (%)	73.8 ± 16.5	79.17 ± 10.7	0.557
AHI, Total PLM index, Total	3.7 ± 5.2 8.8 ± 19.7	4.3 ± 6.5 14.6 ± 22.9	0.859 0.557
Arousal index, Total	11.4 ± 4.7	12.3 ± 8.0	0.974

# Fisher’s exact test. * *p* < 0.05. RBD: REM sleep behavior disorders, UPDRS: Unified Parkinson’s Disease Rating Scale, ISI: Insomnia Severity Index, ESS: Epworth Sleepiness Scale PSQI: Pittsburgh Sleep Quality Index, RBDQ-KR: RBD Questionnaire—Korean version, BDI: Beck Depression Index, BAI: Beck Anxiety Index, MMSE: Mini-Mental Status Examination, SCOPA-AUT: Scales for Outcomes in Parkinson’s Disease for Autonomic Symptoms, KVSS: Korean Version of Sniffin’ Sticks Test, HMR: Heart to mediastinal ratio, RWA: REM sleep Without Atonia, AHI: Apnea—Hypopnea index.

**Table 3 jcm-10-05414-t003:** Comparative analysis within iRBD group based on the cut-off value of delayed HMR.

	≥2.05 (*n* = 4)	<2.05 (*n* = 35)	*p*-Value
Age	63.5 ± 9.1	65.6 ± 6.2	0.831
Sex (male, %)	3 (75.0)	22 (62.9)	0.545 ^#^
UPDRS part III	1.00 ± 2.0	0.2 ± 0.9	0.197
Onset age, year	53.0 ± 13.9	60.2 ± 6.9	0.166
Duration, year Interval of symptoms to PSG, year	13.8 ± 11.0 12.5 ± 10.3	5.6 ± 2.8 3.6 ± 3.0	0.078 0.179
ISI	6.3 ± 7.9	6.0 ± 5.6	0.902
ESS	2.0 ± 2.5	2.1 ± 4.3	0.122
BDI	7.8 ± 11.6	12.5 ± 10.8	0.426
BAI	4.3 ± 7.9	7.5 ± 8.0	0.293
PSQI	5.5 ± 2.5	6.7 ± 3.8	0.644
MMSE	25.5 ± 3.1	27.3 ± 2.3	0.242
RBDQ-KR Factor 1 Factor 2	38.3 ± 20.1 9.8 ± 8.3 29.0 ± 17.2	43.3 ± 18.2 13.3 ± 5.6 30.4 ± 14.8	0.815 0.268 0.865
SCOPA-AUT total	7.0 ± 5.0	9.1 ± 4.7	0.455
Gastrointestinal	2.0 ± 1.6	2.6 ± 2.1	0.659
Urinary	4.3 ± 2.9	4.8 ± 3.0	0.664
Cardiovascular	0.0 ± 0.0	0.6 ± 0.8	0.251
Pupilomotor	0.3 ± 0.5	0.4 ± 0.7	0.920
Thermoreulatory	0.3 ± 0.5	0.3 ± 0.4	1.000
Sexual	0.3 ± 0.5	0.5 ± 0.9	0.822
KVSS test	4.0 ± 3.3	4.2 ± 2.7	0.827
MIBG			
Early HMR	2.6 ± 0.4	1.7 ± 0.2	<0.001 *
Delayed HMR	2.8 ± 0.7	1.5 ± 0.2	<0.001 *
Total sleep time (min)	354.8 ± 93.3	384.0 ± 54.8	0.656
N1 (%)	14.2 ± 5.5	13.6 ± 6.0	0.725
N2 (%)	49.1 ± 11.0	49.8 ± 8.4	0.795
N3 (%)	20.05 ± 3.90	15.1 ± 9.1	0.272
REM (%)	16.6 ± 5.7	21.6 ± 7.3	0.141
Latency to sleep onset	54.5 ± 95.8	12.9 ± 13.1	1.000
Latency to sleep stage 2 (min)	57.4 ± 95.6	19.0 ± 16.1	0.991
Latency to REM sleep stage (min)	114.4 ± 89.9	117.7 ± 69.1	0.945
RWA (%)	9.1 ± 4.3	30.0 ± 22.9	0.011 *
Sleep efficiency (%)	72.9 ± 18.9	79.1 ± 10.5	0.697
AHI, Total PLM index, Total	4.7 ± 5.5 11.0 ± 22.0	4.2 ± 6.4 14.2 ± 22.7	0.733 0.731
Arousal index total	12.7 ± 4.4	12.2 ± 7.9	0.528

# Fisher’s exact test. * *p* < 0.05. RBD: REM sleep behavior disorders, UPDRS: Unified Parkinson’s Disease Rating Scale, ISI: Insomnia Severity Index, ESS: Epworth Sleepiness Scale PSQI: Pittsburgh Sleep Quality Index, RBDQ-KR: RBD Questionnaire—Korean version, BDI: Beck Depression Index, BAI: Beck Anxiety Index, MMSE: Mini-Mental Status Examination, SCOPA-AUT: Scales for Outcomes in Parkinson’s Disease for Autonomic Symptoms, KVSS: Korean Version of Sniffin’ Sticks Test, HMR: Heart to mediastinal ratio, RWA: REM sleep Without Atonia, AHI: Apnea—Hypopnea index.

## Data Availability

The data that support the findings of this study are available from the corresponding author, upon reasonable request.

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
