# Peer review of "Cardiac Autonomic Dysfunction Is Associated with Severity of REM Sleep without Atonia in Isolated REM Sleep Behavior Disorder"

_jcm, 2021, doi:10.3390/jcm10225414_

Round 1

Reviewer 1 Report

This paper is very important. cardiac MIBG is lower even in patients with isolated RBD.

1) How was the relationship between the other autonomic parameters and MIBC findings?

2) Could you show MOCA (cognitive function measure) or olfactory function of the subject patients?

3) Some iRBD patients with normal MIBG finding develop to multiple system atrophy but not to PD. Do you have some comment about this?

Reviewer 2 Report

We have to congratulate the authors for a well written manuscript.

We agree with authors the main limitations of this study the small sample studied,and the lack of matching between control and cases.

I miss the posibiity to include a Dopamine Dat Scan to exclude silence prevalence PD  in cases or controls.

We believe this manuscript deserved to be published as it is original in the correlation of both tests.

Round 2

Reviewer 1 Report

I feel that this paper is well written.